# The Added Value of Serum Random Cortisol and Thyroid Function Tests as Mortality Predictors for Critically Ill Patients: A Prospective Cohort Study

**DOI:** 10.3390/jcm11195929

**Published:** 2022-10-08

**Authors:** Narakorn Muentabutr, Worapaka Manosroi, Nutchanok Niyatiwatchanchai

**Affiliations:** 1Department of Internal Medicine, Faculty of Medicine, Chiang Mai University, Chiang Mai 50200, Thailand; 2Division of Endocrinology, Faculty of Medicine, Chiang Mai University, Chiang Mai 50200, Thailand; 3Division of Pulmonary, Critical Care and Allergy, Faculty of Medicine, Chiang Mai University, Chiang Mai 50200, Thailand

**Keywords:** thyroid function test, random cortisol, critical illness, mortality, predictor

## Abstract

Background: Thyroid hormone and cortisol levels can change during a course of illness. Our study was conducted to assess the ability of the level of these hormones to predict mortality among intensive care unit (ICU) patients. The added predictive value of these hormones with APACHE II scores was also evaluated. Methods: Thyroid hormones and random cortisol levels in adult ICU patients were collected on admission. Multivariate logistic regression analysis was used to assess the relationship between hormone levels and mortality. The added value of the mortality predictive ability was determined by area under the receiver operating characteristic (AuROC). Results: A total of 189 patients were included in the study. Free T3 and serum random cortisol levels were statistically significantly related to ICU mortality with OR 0.51 (0.28, 0.97), *p* = 0.047 and OR 1.02 (1.01, 1.04), *p* < 0.002, respectively. Free T3 and serum random cortisol significantly enhanced the predictive performance of APACHE II scores with an AuROC of 0.656 (non-added value model) versus 0.729 (added value model), *p* = 0.009. The scoring system was created with a total score that ranged from 1 to 14. A score above 7.0 indicated a high mortality rate with a sensitivity of 81.5% and a specificity of 33%. Conclusions: Serum free T3 and cortisol levels are significantly associated with ICU mortality and can enhance the ability of APACHE II scores to predict ICU mortality.

## 1. Introduction

In critically ill patients, proper organ response is a key factor in survival. The endocrine system plays a crucial role in helping the body cope with potentially fatal stressors through the secretion of a variety of different hormones [1]. Alterations of thyroid hormones and cortisol are among the most studied hormones in critically ill patients. During the early hours of a critical illness, the free T3 level rapidly declines, while reverse T3 increases. If the severity of the illness progresses, free T4 and TSH start to decline together with persistently low free T3 [2]. These alterations of thyroid hormones during illness are often known collectively as “non-thyroidal illness syndrome” (NTIS) or “euthyroid sick syndrome”. A several-fold rise of serum cortisol levels has been demonstrated to occur in critical ill patients and can be explained by multiple mechanisms [3]. This response is believed to be essential for survival during severe illness [4]. Other hormones, including the somatotropic axis and gonadal axis, have also been reported to change during critical illness. For the somatotropic axis, low circulating levels of IGF-1 resulting from hepatic GH receptor resistance has been reported. The testosterone levels in males and estradiol in females are also decreased during a period of critical illness [5].

Several studies have attempted to demonstrate an association between thyroid hormone values and mortality. Most of those studies found a significant association between thyroid hormone levels, especially T3, with mortality in ICU patients [6,7]. One of these studies also reported a significant benefit from including T3 in Acute Physiology and Chronic Health Evaluation II (APACHE II) scores, which is a disease severity assessment tool used to predict ICU mortality [7]. Various studies have demonstrated a significant relationship between serum random cortisol and ICU mortality in patients with community-acquired pneumonia and patients with COVID-19 infection as well as septic and traumatic ICU patients [8,9,10,11].

Disease severity assessment tools have been developed to assist physicians in selecting treatment based on an individual’s disease severity [12]. Among the disease severity assessment tools, APACHE II is one of the most widely recognized tools for assessing critically ill patients. APACHE II scores have been validated for prediction of ICU mortality, but that classification system needs to be updated over time to remain effective due to changes in incidence rates of diseases and improvements in treatment efficacy [13]. However, at present, there have been no studies which include both thyroid function tests and random cortisol to predict ICU mortality. As a result, evidence of the value of combining these hormone level values with APACHE II scores to enhance the ability to predict ICU mortality is still lacking. The present study was conducted to evaluate the association of thyroid hormone and cortisol levels within the first day of ICU admission with ICU mortality as well as to determine the appropriate cut-off levels for the ease-of-use in clinical practice. Other objectives were to assess whether these hormone levels add value to APACHE II scores in terms of predicting ICU mortality and to create scoring system by incorporating these hormones with APACHE II scores.

## 2. Materials and Methods

This prospective observational study was conducted in adult patients admitted to the medical ICU at Maharaj Nakorn Chiang Mai Hospital, which is a tertiary care hospital in Thailand. The recruitment period was from January 2021 to November 2021. The study was approved by the Faculty of Medicine, Chiang Mai University ethical committee (study number: MED-2563-07388). The inclusion criteria were all patients admitted to the medical ICU during the study period due to acute medical indications who were aged 18 years or over. The exclusion criteria included: (1) current diagnosis of diseases related to thyroid dysfunctions, (2) current diagnosis of adrenal insufficiency, (3) received medications prior to admission that affect thyroid hormones or cortisol levels, (4) received oral, injectable, topical or inhaled steroids during the inclusion selection process or the use of such steroids for more than 1 week and discontinued less than 1 month prior to admission, (5) a history of pituitary gland disorders, e.g., pituitary surgery or radiation, (6) pregnancy and (7) inability to obtain a complete physical examination and laboratory results within the first 24 h of ICU admission. Written informed consent was obtained from the patient or a relative prior to recruitment. Patients’ laboratory investigations, including free T3 level, free T4 level, TSH level, random cortisol levels as well as other laboratory investigations, comprehensive history taking, and physical examination were completed within the first 24 h of ICU admission. Thyroid function tests were collected between 6 and 9 a.m. in the morning, while serum cortisol levels were collected randomly. The patients’ demographic data and laboratory investigation results were used to calculate APACHE II scores [14]. The APACHE II score comprises 12 physiological and two disease-related variables, which are age, underlying diseases, PaO2, body temperature, mean arterial pressure, blood pH, heart rate, respiratory rate, serum sodium, potassium, creatinine, hematocrit, white blood cell count and Glasgow coma score. The score ranges from 0 to 71. The higher the score, the more severe the disease and the higher the risk of death. The outcome of interest was ICU mortality. Included patients who were discharged from the hospital were not eligible to be recruited again at a later date. All study procedures were conducted in accordance with the ethical standards of the responsible committees on human experimentation (institutional and national) and with the Declaration of Helsinki 1975 as revised in 2008.

### 2.1. Laboratory Assays

Serum FT3 (normal reference range 2.04–4.40 pg/mL), FT4 (normal reference range 0.93–1.71 ng/dL), and TSH (normal reference range 0.27–4.2 μIU/mL) were measured by electrochemiluminescence assay (ECLIA) (Elecsys^®^ FT3 III, FT4 III, TSH assay, Roche Diagnostics GmbH, Mannheim, Germany) with intra- and inter-assay variation of 1.9–8.2%. Serum cortisol levels were measured by electrochemiluminescence assay (ECLIA) (Elecsys^®^ Cortisol II assay, Roche Diagnostics GmbH, Mannheim, Germany). The intra- and inter-assay coefficients of variation for serum cortisol were <10%.

### 2.2. Statistical Analysis

Data were analyzed using STATA version 16 (StataCorp, Lakeway, TX, USA). Categorical variables are reported as frequencies and percentages. Continuous variables are presented as means and standard deviations (SD). For inferential statistics, chi-squared or Fisher’s exact test were used for categorical data. Continuous data were analyzed using the independent t-test for normally distributed data or the Mann–Whitney U test for non-normally distributed data. Univariate and multivariate logistic regression were used to determine the association between hormonal tests and ICU mortality. Only variables which had a *p*-value < 0.2 from univariate analysis were chosen for multivariate analysis. Multivariate analysis was adjusted for age, sex, serum albumin, mechanical ventilators, inotropic medications and CRRT used. Results are reported as odds ratios (ORs) with 95% confidence intervals. The accuracy of the proposed cut-off levels for hormonal investigations are presented with sensitivity, specificity, positive predictive value (PPV), negative predictive value (NPV) and area under the receiver operating characteristic (AuROC) curves. AuROC curves were also employed to assess the performance of variables in predicting ICU mortality.

Item scores were calculated by the transformation of the regression coefficient. The coefficient of each level for each factor was divided by the smallest coefficient of the model and rounded to the nearest 0.5. Item scores were then added together to calculate a total score. The total scores were then divided into 2 risk levels: groups at a low risk and those at a high risk of mortality rate based on the cut-off which provided the highest sensitivity.

A two-sided *p*-value < 0.05 was considered statistically significant. Sample size was calculated using a STATA program and was based on a study from Gutch et al. The AuROC of the APACHE II model was 0.824, and the ratio of non-survivors to survivors was 81 to 189 [15]. We presumed the AuROC of the added-on thyroid function test and random cortisol with APACHE II score to be 0.900. Type I error was determined to be 0.05, and type II error was 0.20. Therefore, at least 154 patients with 66 non-survivors and 88 survivors needed to be included.

## 3. Results

A total of 690 adult patients admitted to the medical ICU were evaluated and recruited into the study. Of these, 151 patients met the exclusion criteria and were excluded. An additional 350 patients were excluded because their laboratory investigations could not be acquired within the first 24 h of admission. Therefore, a total of 189 patients were recruited into the study. The study flow is shown in Figure 1.

### 3.1. Baseline Characteristics

Baseline characteristics are shown in Table 1. The included populations were predominantly male (56.1%, *n* = 106/189). The mean age of the recruited patients was 68.22 ± 16.76 years. Chronic organ failure, e.g., cirrhosis, chronic heart failure and chronic kidney diseases (27.5%, *n* = 52/189) were the most common underlying diseases followed by metastatic cancer (15.9%, 30/189). The most common final diagnosis was pneumonia (29.6%, *n* = 56/189) followed by sepsis and septic shock (21.2%, *n* = 40/189). Most of the patients 89.4 % (*n* = 169/189) were given mechanical ventilation on the first day of ICU admission, while 42.9% (*n* = 81/189) were administered inotropic medications on admission and 43.4% (*n* = 82/289) were on continuous renal replacement therapy (CRRT). The mean duration of ICU admission was 11.3 ± 13.5 days, and 43.1% of patients were admitted for longer than 7 days. The ICU mortality among the patients was 48.6% (92/189).

There was a difference in the incidence of underlying diseases of survivors and non-survivors (Table 1). ICU admission duration was significantly longer for non-survivors than for survivors. Respiratory rate on admission, A-a gradient, serum bilirubin and white blood cell count were significantly higher in non-survivors than in survivors. Serum albumin, cholesterol and platelet counts were significantly lower in non-survivors than in the survivor group. APACHE II scores were higher among patients who did not survive.

Laboratory investigations found that the mean free T3 level was significantly lower in non-survivors (1.27 ± 0.53 pg/mL) compared with survivors (1.46 ± 0.47 pg/mL), *p* = 0.007. The mean random cortisol level in non-survivors was significantly higher (36.19 ± 31.54 µg/dL) than in survivors (21.71 ± 14.34 µg/dL), *p* <0.001. TSH and free T4 were not significantly different between the two groups.

### 3.2. Associations of Thyroid Hormones and Cortisol with ICU Mortality

Univariate analysis showed that free T3 and serum random cortisol level were significantly correlated with ICU mortality with OR 0.45 (95%CI 0.24–0.82), *p* = 0.009 and OR 1.03 (95%CI 1.01–1.05), *p* < 0.001, respectively. Free T4 and TSH levels were not significantly correlated with ICU mortality. Multivariate logistic regression analysis adjusted for age, sex, albumin, mechanical ventilators, inotropic medications and CRRT used found that free T3 and random cortisol levels were significantly associated to ICU mortality. The lower the free T3 level, the higher the ICU mortality risk (OR 0.51 (95%CI 0.28–0.97), *p* = 0.047) and the higher the serum random cortisol, the higher the ICU mortality risk (OR 1.02 (95%CI 1.01–1.04), *p* < 0.002) (Table 2).

The proposed cut-off levels for free T3 and random cortisol levels to predict ICU mortality are shown in Table 3. The cut-off level for free T3 to help predict ICU mortality which showed high sensitivity without excessively low specificity was <1.7 pg/mL with a sensitivity of 81.5% and specificity of 26.8%. Using these cut-off levels, 9% of the patients (*n* = 17/189) who showed false negative results, i.e., those who had free T3 > 1.7 pg/mL, did not survive. The cut-off random cortisol level to predict ICU mortality was >15 µg/dL with a sensitivity of 80.4% and specificity of 35.1%. Using this cut-off level, 9.5% (18/189) of the patients in this study showed false negative results.

### 3.3. Added Value of Free T3 and Random Cortisol with APACHE II Score

The AuROC of APACHE II scores, free T3, random cortisol and the add-on value of those hormones in combination with APACHE II scores are summarized in Table 4. Among the hormonal parameters, random cortisol had the highest AuROC (0.725, 95%CI 0.654–0.796) followed by free T3 (0.708, 95%CI 0.632–0.782). The APACHE II scores showed the lowest AuROC (0.656, 95%CI 0.579–0.734). The added value model of random cortisol and APACHE II scores had a significantly higher AuROC compared with the APACHE II model alone (AuROC 0.728 versus 0.656, *p* = 0.010), but there was no observed significant difference in the free T3 added value model (AuROC 0.707 versus 0.656, *p* = 0.280). If serum random cortisol and free T3 were added to the APACHE II model, a significantly higher AuROC than in the APACHE II model alone was demonstrated (AuROC 0.728 versus 0.656, *p* = 0.009). The figures of AuROC are as shown in Figure 2.

### 3.4. Scoring System

A predictive scoring system was created to predict the mortality rate. The transformed scores ranged from 1.0 to 7.0 with a total score of 14. The scoring system is shown in Table 5. The total scores were classified into two groups: a low-risk group (scores 1–7.0) and a high-risk group (scores above 7.0). This cut-off had a sensitivity and a specificity of 81.5% and 33.0%, respectively.

## 4. Discussion

This prospective study demonstrated that free T3 and serum random cortisol can help predict ICU mortality and that incorporating both hormones could significantly enhance the predictive ability of APACHE II scores. These hormonal changes can be used together with APACHE II scores to guide calculation of the mortality risk of ICU patients. For ease of use in general practice, appropriate cut-off levels for each of the hormones and scoring system were also proposed in this study. The association of these hormonal changes with ICU mortality risk was consistent with previous studies, including one meta-analysis [6,11,16]. Even though a study which showed a significant relationship of cortisol with ICU mortality had AuROC values comparable to the present study, that study used serum morning cortisol rather than serum random cortisol as in the present study, and a different cortisol assay was employed [11].

The associations of free T3 and cortisol with ICU mortality have been reported in previous studies. One study reported that only serum cortisol response after an ACTH stimulation test had a significant relationship with ICU mortality, while morning cortisol samples had no significant association [17]. Other studies reported that free T4 had a significant relationship with ICU mortality, which was a relationship that was not found in the present study [6,7]. However, their studies reported a significant association of free T4 and mortality based only on univariate analysis. Therefore, some confounders might not be adjusted, and their results might not be conclusively concluded based on univariate analysis. Previous reports of the added value of free T3 to APACHE II scores were similar to the present study [6]. To the best of our knowledge, there have been no studies that demonstrate the added value of random cortisol and both random cortisol with free T3 to APACHE II scores.

Both hormonal changes and their relationship to ICU mortality could be explained by the underpinning physiology. The thyroid hormone has been hypothesized to be associated with stress. A condition known as NTIS is a condition in which thyroid hormone levels change due to physiologic response to stress. These changes included low free T3, low free T4, increased plasma reverse T3 (rT3) and normal range or slightly decreased TSH levels [18]. It is still uncertain whether NTIS is a normal beneficial physiologic response. NTIS is believed to be an energy reduction mechanism that improves the ability to survive under stress, as the same hormonal changes can be found during fasting [18]. The mechanisms of NTIS are a decrease in hypothalamic–pituitary–thyroid axis activation and decreased deiodinases type 1 and 2, leading to a reduction in the conversion of thyroxine (T4) to triiodothyronine (T3) and increased hepatocyte deiodinases type 3, resulting in the conversion of T3 to rT3 [19]. In prolonged critical illness, levels of all free T3, free T4 and TSH are low [20]. There is currently no evidence-based consensus regarding the treatment of NTIS in critically ill patients. Focusing on the underlying causes of the critical illness is still the main modality of treatment in critically ill patients.

Multiple mechanisms have been proposed to explain serum cortisol alterations during critical illness. One report showed that the clearance of serum cortisol was reduced during critical illness as result of a reduction in cortisol-metabolizing enzymes such as α-ring reductases in the liver and 11β-hydroxysteroid dehydrogenase 2 in the kidney. Additionally, a decline of cortisol-binding globulin (CBG), the main cortisol carrier in blood circulation, has been shown to be markedly decreased during critical illness and to lead to an increase in the unbound or free form of cortisol. This change coincides with decreased cortisol metabolism, resulting in substantially increased free and total forms of cortisol availability [21,22]. The theory that the increased activation of the hypothalamic–pituitary–adrenal axis (HPA) leads to a higher production of serum cortisol from adrenal glands during critical illness is still open to debate, as there have been multiple studies reporting that the ACTH level was not found to be higher despite an increased serum cortisol level [22,23]. Based on all of the above, it can be presumed that a higher stress response, i.e., a higher the degree of illness, can lead to higher levels of serum cortisol. The increased levels of cortisol in response to critical illness are believed to be beneficial in various ways including increased cardiac contractility, reduction in heart rate, maintenance of vascular tone, and maintenance of vascular volume via the interaction of angiotensin, vasopressin and atrial natriuretic peptide [24].

However, the causal association between decreased free T3 and increased serum cortisol with ICU mortality cannot be demonstrated from this study. If low free T3 was the cause of increased ICU mortality, the intervention aimed to normalized free T3 level should help improved mortality rate. To date, this question cannot be answered based on small numbers of randomized controlled trials. These trials included a heterogeneous NTIS population in terms of disease severity and age group. In addition, a variety of thyroid hormone treatments (T3 and T4) were employed in these trials. The results of thyroid hormone treatment in critical illness have been mostly negative in terms of clinical benefit [25,26,27,28]. Therefore, NTIS could not be explained solely from decreased serum thyroid hormone concentration in plasma. It could be presumed that thyroid hormone concentration in plasma might not be equal to thyroid hormone concentration in tissue or organs. Regarding serum cortisol and ICU mortality, to the best of our knowledge, no research has been conducted on the effects of lowering serum cortisol and ICU mortality. The question regarding the causal relationship between these hormonal changes and ICU mortality can only be answered by a future large randomized multicenter controlled trial. APACHE II scores are the commonly used parameter to estimate ICU mortality, incorporating multiple laboratory investigations, patients’ health status and physiologic variables. Primarily, serum random cortisol and free T3 are not normally included in those scores. This study added serum cortisol and free T3 to APACHE II scores to determine if that combination improved the predictive ability of the model. Combining the value of serum cortisol and free T3 with APACHE II scores did show a significant improvement in mortality prediction accuracy. However, the AuROC of APACHE II scores in the present study (AuROC 0.656) differed from other studies where AuROC varied from 0.56 to 0.88 [6,29]. This could be the result of differences in the population included in the present cohort compared to other cohorts. In addition, since the development of the initial APACHE II in 1981, it has needed to be modified and updated over time [30]. Combining biochemical values such as free T3 and random cortisol, values which are readily available and easy to obtain in many institutions, can help improve the predictive ability of the APACHE II score. Moreover, the cut-off levels of serum free T3 (<1.7 pg/mL) and serum random cortisol (>15 µg/dL) developed in the present study provide a high sensitivity of >80% in the prediction of ICU mortality. As ICU mortality is a lethal condition, a cut-off level with high sensitivity was chosen to reduce the number of false negative results.

The present study had multiple strengths. First, the sample size was large enough to adequately evaluate the outcome of interest. Second, appropriate cut-off levels of the hormonal predictors were determined, which are levels that are easy to use and practical in actual medical practice. Third, the findings can be explained by underpinning physiology and are thus less likely to have occurred by chance. The prediction of mortality in ICUs plays an important role in patient care and resource allocation. The early identification and management of patients at risk of death is associated with lower mortality rates. Our results are a potential starting point for the use of multiple endocrine laboratory results, particularly random cortisol and free T3, together with APACHE II scores, to identify patients at a high risk of death and to lead to more appropriate treatment.

The present study also had multiple limitations. First, the study did not exclude patients with unknown status of pre-existing thyroid diseases, adrenal diseases or pituitary gland diseases. Due to the complexity of diagnosing these diseases, many patients remained undiagnosed during their stay in the ICU. The unknown status of pre-existing hormonal disorders may have introduced inaccuracies in the interpretation of results. Second, the patients included in the study may not be representative of patients with critical illnesses in general. The study population was predominantly elderly and included a high proportion of patients who were mechanically ventilated, which is a mixed-diagnosis group that would be expected to have a very high mortality rate. Third, the proposed cut-off levels for free T3 and random cortisol were based on specific laboratory assays. Prior to general application in other institutions, these cut-off levels and the scoring system will need to be validated. Fourth, as there was evidence of decreased levels of CBG in ICU patients during critical illness, endeavoring to obtain serum random cortisol in this study rather than the free form of cortisol may not have been realistic. Fifth, the sample size in this study was relatively small. However, it does offer sufficient statistical power (>80%) for the analysis. Lastly, the number of patients who were excluded due to laboratory investigations that could not be acquired within the first 24 h of admission was high. However, the mortality rate in these patients was similar to that in the included patients (46.8% versus 48.6%).

## 5. Conclusions

Serum free T3 and cortisol levels are significantly associated with ICU mortality. Free T3 and serum random cortisol levels combined with APACHE II scores can enhance the ability to predict ICU mortality. These findings can be employed as a guide to help clinicians more accurately predict mortality risk in ICU patients and thus help determine appropriate treatment. Further external validation of these findings and the development of predictive scoring system is warranted.

## Figures and Tables

**Figure 1 jcm-11-05929-f001:**
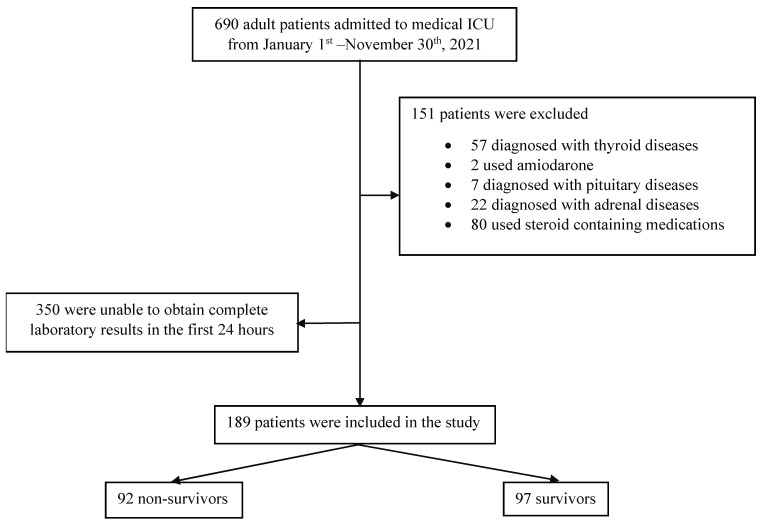
Study flow.

**Figure 2 jcm-11-05929-f002:**
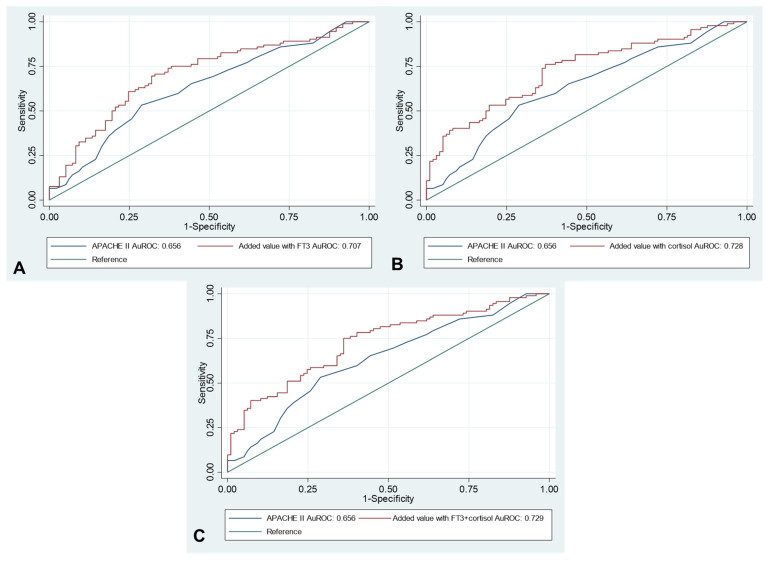
Added value of free T3 (**A**) serum random cortisol (**B**) and free T3 plus serum random cortisol (**C**) on APACHE II score to predict intensive care unit mortality presented by area under receiver operating characteristic curve.

**Table 1 jcm-11-05929-t001:** Baseline characteristics.

Characteristics	Non-Survivor (*n* = 92)	Survivor (*n* = 97)	*p*-Value
Male, *n* (%)	54 (58.7)	52 (53.6)	0.481
Age, mean ± SD (year)	70.1 ± 14.8	66.5 ± 18.3	0.139
Weight, mean ± SD (kg)	54.1 ± 10.9	56.7 ± 15.4	0.180
Underlying diseases, *n* (%)			
Metastatic cancer	20 (21.7)	10 (10.3)	0.009
AIDS	1 (1.1)	0 (0)	
Hematologic malignancy	11 (11.9)	3 (3.1)	
Chronic heart failure or cirrhosis or chronic renal failure or chronic lung disease	26 (28.3)	26 (26.8)	
Others	14 (15.2)	6 (6.2)	
Final diagnosis, *n* (%)			
Pneumonia	24 (26.0)	32 (32.9)	0.227
Sepsis or septic shock	27 (29.3)	13 (13.4)	
DKA or HHS	0 (0)	7 (7.2)	
COPD with acute exacerbation	4 (4.3)	2 (2.1)	
Acute heart failure	10 (10.9)	15 (15.5)	
Stroke	0 (0)	2 (2.1)	
Gastrointestinal tract hemorrhage	1 (1.1)	6 (6.2)	
Others	26 (28.4)	20 (20.6)	
ICU duration, mean ± SD (day)	13.3 ± 16.0	9.4 ± 10.3	0.043
Mechanical ventilation on admission, *n* (%)	86 (93.5)	83 (85.6)	0.077
Inotropic medication used on admission, *n* (%)	45 (48.9)	36 (37.1)	0.101
Continuous renal replacement therapy, *n* (%)	46 (50.0)	36 (37.1)	0.07
Vital status on admission, mean ± SD			
Systolic blood pressure (mmHg)	116.6 ± 24.0	110.3 ± 23.7	0.279
Diastolic blood pressure (mmHg)	65.5 ± 15.0	65.1 ± 15.0	0.853
Mean arterial pressure (mmHg)	82.5 ± 15.6	83.5 ± 15.9	0.669
Body temperature (°C)	37.1 ± 1.5	37.2 ± 1.0	0.717
Pulse rate (per minute)	101.2 ± 24.6	95.4 ± 24.7	0.109
Respiratory rate (per minute)	22.8 ± 5.9	20.7 ± 5.7	0.013
Arterial blood gas, mean ± SD			
pH	7.4 ± 0.1	7.4 ± 0.1	0.447
PaO2 (mmHg)	118.6 ± 68.4	124.4 ± 66.1	0.554
PaCO2 (mmHg)	32.9 ± 11.4	34.4 ± 10.6	0.367
A-a gradient, mean ± SD	214.7 ± 146.1	150.9 ± 116.5	0.001
Blood chemistry, mean ± SD			
BUN (mg/dL)	41.45 ± 29.9	36.8 ± 31.3	0.297
Creatinine (mg/dL)	2.5 ± 0.6	2.7 ± 0.9	0.840
Sodium (mmol/L)	136.4 ± 6.9	137.1 ± 6.1	0.495
Potassium (mmol/L)	3.9 ± 0.8	4.0 ± 0.8	0.600
Bicarbonate (mmol/L)	19.9 ± 6.3	20.6 ± 5.5	0.470
Serum albumin (g/dL)	2.7 ± 0.6	3.0 ± 0.6	<0.001
Cholesterol (mg/dL)	107.3 ±46.9	124.0 ± 46.6	0.015
Total bilirubin (mg/dL)	2.4 ± 0.9	1.2 ± 0.5	0.025
Complete blood count, mean ± SD			
Hematocrit (%)	28.1 ± 6.5	29.3 ± 8.9	0.316
White blood cell count (cell/mcL)	16,838.1 ± 5972.0	12,697.8 ± 1203.6	0.031
Platelet count (cell/mcL)	182,326.1 ± 25,841.0	236,123.7 ± 24,430.8	0.003
APACHE II score, mean ± SD	19.6 ± 7.6	16.1 ± 6.7	<0.001
TSH (µIU/mL), mean ± SD	3.4 ± 6.1	3.1 ± 9.1	0.797
Free T4 (ng/dL), mean ± SD	1.1 ± 0.5	1.2 ± 0.5	0.118
Free T3 (pg/mL), mean ± SD	1.3 ± 0.5	1.5 ± 0.5	0.007
Random cortisol (µg/dL), mean ± SD	36.2 ± 31.5	21.7 ± 14.3	<0.001

SD: Standard deviation, AIDS: Acquired immunodeficiency syndrome, DKA; Diabetes ketoacidosis, HHS: Hyperglycemic hyperosmolar syndrome, COPD: Chronic obstructive pulmonary disease, TSH: Thyroid-stimulating hormone.

**Table 2 jcm-11-05929-t002:** Univariate and multivariate analysis of relationship of thyroid function tests and cortisol with ICU mortality.

Predictors	Univariate Analysis			Multivariate Analysis *		
	ORs	95% CI	*p*-Value	ORs	95% CI	*p*-Value
TSH	1.01	0.97–1.04	0.970	-	-	-
Free T4	0.59	0.31–1.15	0.125	0.69	0.35–1.38	0.208
Free T3	0.45	0.24–0.82	0.009	0.51	0.28–0.97	0.047
Cortisol	1.03	1.01–1.05	<0.001	1.02	1.01–1.04	<0.002

* Only variables with *p* < 0.20 based on univariate analysis were chosen for multivariate analysis and were adjusted for age, sex, inotropic medication, mechanical ventilator and continuous renal replacement therapy used.

**Table 3 jcm-11-05929-t003:** Proposed cut-off levels for free T3 and random cortisol for predicting intensive care unit mortality.

	Cut-Off	Sensitivity %	Specificity %	PPV %	NPV %	TP (*n*)	FN (*n*)	FP (*n*)	TN (*n*)	AuROC
Free T3	<1.5 pg/mL	69.6	43.3	53.8	60.0	64	28	55	42	0.534
	<1.6 pg/mL	73.9	34.0	51.5	57.9	68	24	64	33	0.539
	<1.7 pg/mL	81.5	26.8	51.4	60.5	75	17	71	26	0.541
	<1.8 pg/mL	84.8	19.6	50.0	57.6	78	14	78	19	0.521
	<1.9 pg/mL	90.2	12.4	49.4	57.1	83	9	85	12	0.512
Random cortisol	>13 µg/dL	85.9	23.7	51.6	63.9	79	13	74	23	0.547
	>14 µg/dL	83.7	29.9	53.1	65.9	77	15	68	29	0.568
	>15 µg/dL	80.4	35.1	54.0	65.4	74	18	63	34	0.577
	>16 µg/dL	78.3	37.1	54.1	64.3	72	20	61	36	0.576
	>17 µg/dL	73.9	41.2	54.4	62.5	68	24	57	40	0.575

**Table 4 jcm-11-05929-t004:** Performance of predictors presented as area under ROC curve.

Predictors	AuROC	95% Conf. Interval	*p*-Value
APACHEII score	0.656	0.579–0.734	-
Free T3	0.708	0.632–0.782	-
Random cortisol	0.725	0.654–0.796	-
APACHE II score + free T3	0.707	0.632–0.781	0.280 *
APACHE II score + random cortisol	0.728	0.656–0.799	0.010 *
APACHE II score + free T3 + random cortisol	0.729	0.658–0.801	0.009 *

* Compared to APACHE II score model.

**Table 5 jcm-11-05929-t005:** Scoring system to predict ICU mortality.

	Coefficient	Transformed Coefficient	Assigned Score
APACHE II >30 *			
-Yes			
-No	0.64	5.81	6
Random cortisol >15 µg/dL			
-Yes			
-No	0.78	7.09	7
FT3 <1.7 pg/mL			
-Yes			
-No	0.11	1	1
Total score			14

* Cut-off based on median value.

## Data Availability

The data that support the findings of this study are available upon request from the authors.

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
