# Peer review of "The Added Value of Serum Random Cortisol and Thyroid Function Tests as Mortality Predictors for Critically Ill Patients: A Prospective Cohort Study"

_jcm, 2022, doi:10.3390/jcm11195929_

Round 1

Reviewer 1 Report

Narakorn Muentabutr et al informed us, that random serum levels of FT3 and cortisol could predict ICU mortality and the incorporation of these parameters to the APACHE II score increased its predictive value. The association between thyroid hormones with mortality in ICU patients has been extensively studied in the literature and the present study does not have any novelty. The specificity of the proposed cut-offs for both parameters is very low thus the PPV is also very low, so they are useless. Also, the authors showed that the addition of these parameters improves the AuROC for the APACHE II score but in clinical practice they did not provide us with any calculator and a score highly suggestive for death.

Minor Remarks:

page 6; lines 173 & 176: the percentages of false negative results are at variance with sensitivity

Reviewer 2 Report

This study looks at the prognostic value of serum fT3 and morning cortisol taken within 24 hours of ITU admission as a predictor of mortality. These data were combined with the APACHEII score with the intention of increasing its predictive power.  Whilst the association of low fT3 and high cortisol with ITU mortality is well reported - this is a well designed and reported study and which adds further evidence around the value of these parameters. 

My only concern with the study is the number of patients  that were excluded as no sample was taken within 24 hours.   This is a potential confounder. I think if we had some idea of the severity of illness in the group that didn't have fT3 and cortisol measured this would add weight to this study.  Are APACHE II scores or baseline characteristics available for the non-tested group?  I think we need some reassurance that this group are reasonably similar to the group tested. 

Some authors have suggested that serum albumin is as accurate as APACHE II for predicting mortality in critically ill patients; in this cohort albumin would seem to be a promising marker based on p-value (as is platelet count).   I wonder if this could be included in the analysis? Low Albumin would be expected to cause low  serum (total) cortisol so raised cortisol may be of greater significance in hypoalbuminemia.

Minor points - the authors use the term random cortisol but state "For thyroid function tests and serum cortisol, they were collected in the morning 83 between 6AM and 9AM." in the methods.

It may be worth stating briefly the algorithm and  parameters used to calculate the APACEII score rather than citing the original reference.

Round 2

Reviewer 1 Report

The ms has much improved. The creation of the scoring system which incorporates FT3 and cortisol into APACHEII helps clinicians to make a decision, mainly with negative predictive value. However, the low NPV and PPV of the proposed cut-off levels for serum FT3 and cortisol for predicting mortality in ICU patients limits the value of data.

Minor remarks:

page 6; lines 187 & 190: Still, the percentages of false negative results are at variance with the proposed sensitivity.  The formula is “Sensitivity=TP/TP+FN”, so if the authors reported a sensitivity of 81,5% for the cut-off value <1.7 pg/mL for the FT3, that means, from those ICU patients with FT3> 1.7 pg/mL, 18,5% will die (False Negatives) and not 9%. The same for cortisol. Something is wrong.
